# Isotopic evidence for the formation of the Moon in a canonical giant impact

Sune G. Nielsen [1,2 ✉], David V. Bekaert[1] & Maureen Auro[1]

Isotopic measurements of lunar and terrestrial rocks have revealed that, unlike any other body in the solar system, the Moon is indistinguishable from the Earth for nearly every isotopic system. This observation, however, contradicts predictions by the standard model for the origin of the Moon, the canonical giant impact. Here we show that the vanadium isotopic composition of the Moon is offset from that of the bulk silicate Earth by 0.18 ± 0.04 parts per thousand towards the chondritic value. This offset most likely results from isotope fractionation on proto-Earth during the main stage of terrestrial core formation (pre-giant impact), followed by a canonical giant impact where ~80% of the Moon originates from the impactor of chondritic composition. Our data refute the possibility of post-giant impact equilibration between the Earth and Moon, and implies that the impactor and proto-Earth mainly accreted from a common isotopic reservoir in the inner solar system.

[1] NIRVANA laboratories, Woods Hole Oceanographic Institution, Woods Hole, MA, USA. [2] Department of Geology and Geophysics, Woods Hole Oceanographic Institution, Woods Hole, MA, USA. ✉email: snielsen@whoi.edu

There is a general consensus that the last major event in Earth's accretion corresponded to the Moon-forming giant impact[1], but considerable uncertainty remains regarding the exact conditions of this episode[2]. Canonical giant impact models that reproduce the Earth-Moon system require a roughly Mars-sized impactor (called Theia) to have collided with proto-Earth after ~90% accretion was complete[3,4]. These models all result in the Moon predominantly deriving from the impactor, which implies that the Earth and Moon should exhibit distinct isotopic compositions if Theia and proto-Earth formed from different isotopic reservoirs in the solar system. However, extensive isotopic analyses have revealed undetectable or limited difference between the Earth and Moon for elements such as O, Ti and Cr[5–7], which all exhibit large variations among chondrites and differentiated meteorites[6,8,9]. These observations could be explained if (i) the initial dynamic simulations were not capturing the actual geometry of the giant impact itself[10,11], (ii) post-giant impact equilibration between the Earth and Moon materials led to their isotopic homogenization[2,12], or (iii) the impactor and proto-Earth mainly accreted from a common isotopic reservoir (best represented by enstatite chondrites)[13].

Recent studies of V isotope ($^{50}$V and $^{51}$V) variations in chondrites and terrestrial rocks have revealed that the bulk silicate Earth (BSE) is uniformly enriched in $^{51}$V ($\delta^{51}V_{BSE} = -0.856 \pm 0.020‰$; $n = 76$, 2SE; where $\delta^{51}V_{sample} = ((^{51}V/^{50}V)_{sample}/(^{51}V/^{50}V)_{AA} - 1) \times 1000$, with AA corresponding to the Alfa Aesar standard; see Methods section) relative to average chondrites ($\delta^{51}V = -1.089 \pm 0.029‰$, $n = 14$, 2SE), by $\Delta^{51}V_{BSE\text{-}chondrites} = 0.233 \pm 0.037‰$ (2SE)[14]. It has been proposed that V isotope variations in bulk carbonaceous could have a nucleosynthetic origin[15], but subsequently it was found that all V isotope variations in bulk chondrites can be accounted for by recent production of $^{50}$V by GCR spallation processes[16]. The invariant V isotope composition of all chondrites implies that nucleosynthetic V isotope anomalies must be very small and cannot induce planetary scale V isotope heterogeneity. Early solar system irradiation may also have induced production of $^{50}$V by spallation reactions, which has been inferred for some CAIs[17], but given the uniform bulk V isotope compositions in chondrites with highly variable CAI abundances such a process is also unlikely to account for planetary scale V isotope variations. Collectively, these arguments imply that V isotope variations among terrestrial planets do not reflect differences in the composition of their accretionary materials, but are the result of planetary differentiation processes[14]. Although it is currently not certain why the silicate Earth and chondrites are distinct in terms of V isotopes[15], the most likely scenario is that core formation processes prior to the giant impact caused V isotope fractionation that resulted in the BSE having a heavy V isotope composition relative to chondrites[14]. This scenario is supported by the recent observation that the V isotope composition of the bulk silicate mars (BSM) is also enriched in $^{51}$V relative to chondrites ($\Delta^{51}V_{BSM\text{-}chondrites} = 0.067 \pm 0.042‰$[14]). Metal-silicate V isotope fractionation factors required to explain the observed isotope offsets between planetary (BSE, BSM) and chondritic compositions are in good agreement with each other[14], therefore providing empirical support for systematic V isotope fractionation during high pressure-high temperature planetary differentiation, before the Moon-forming event.

Considering the V isotope difference between BSE and chondrites, it follows that if bulk Theia was composed primarily of chondritic material then it should have been significantly lighter than the silicate proto-Earth. As a consequence, the canonical dynamic simulations of the giant impact[3,4] would imply that the Moon is characterized by a V isotope composition intermediate between Earth and chondrites. On the other hand, alternative giant impact geometries[10,11] and post-impact Earth-Moon equilibration scenarios[2,12] would result in largely indistinguishable V isotope compositions for Earth and the Moon, even if Theia and proto-Earth were initially different in terms of $\delta^{51}$V.

Here we show that the Earth and Moon indeed exhibit different V isotope compositions, which strongly implies that the canonical giant impact offers the best explanation for the formation of the Moon.

## Results and discussion

**Lunar V isotope data and correction for cosmic ray exposure.** Recent analyses of lunar rocks have found that many samples exhibit $\delta^{51}$V significantly lighter than both Earth and chondrites[16]. Based on a strong correlation between V isotopes and cosmic ray exposure (CRE) ages, this observation was interpreted to reflect production of $^{50}$V due to galactic cosmic ray (GCR) interaction with primarily Fe atoms (see Supplementary Note 1) at the surface of the Moon[16]. These authors, however, concluded that the Earth and Moon have indistinguishable V isotope compositions due to the large uncertainties on their V isotopic measurements and the large GCR effects on most of their lunar samples. Here, we have analyzed one lunar soil and three lunar basalts with confirmed substantial GCR effects (samples 10084, 15495, 15556, and 70215; GCR exposure ages >100 Myr, Supplementary Table 1) and these all reveal variably light V isotope compositions that follow the previously found relationship between V isotopes and CRE ages (Fig. 1; Supplementary Table 1). In addition, we present new V isotope data for five Apollo mission lunar rocks that have been shown to record very young GCR exposure ages (samples 12004, 74255, 68815, 68115, and 14321; GCR exposure ages between 2 and 49 Myr, Supplementary Table 1). We also analyzed one recently excavated lunar meteorite (LAP02205) that has a very young GCR exposure age (~4 Myr; Supplementary Table 1). These six samples reveal very limited V isotope variation and an uncorrected average $\delta^{51}V = -1.077 \pm 0.039‰$ (2SE), suggesting that this value closely resembles lunar rocks prior to irradiation. When we combine our new V isotope data for lunar samples with previously reported results, we obtain a strong correlation with CRE ages (Fig. 1). The y-intercept of the best-fit line to these data corresponds to

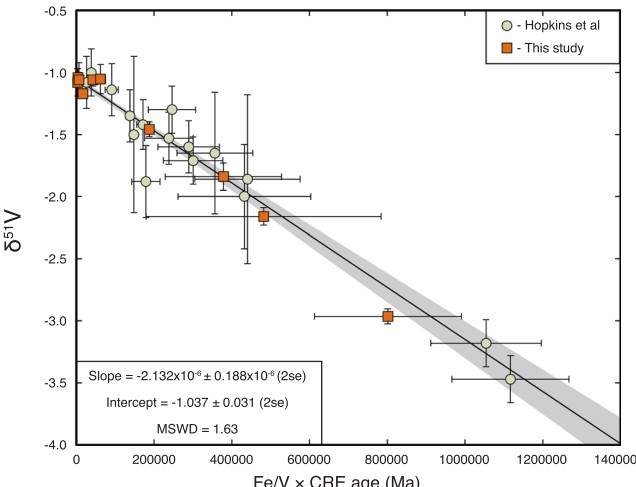

**Fig. 1 Vanadium isotope compositions of lunar samples plotted against their Fe/V-scaled CRE ages.** In this study, Fe/V ratios of all samples were measured on minor splits of the dissolved samples that were processed for V isotope measurements. The best-fit line and gray 2SE envelope through all the data, calculated by taking into account 2SE errors in both x and y, has a y-intercept of $\delta^{51}V_{Moon} = -1.037 \pm 0.031$ (2SE). This value represents our best estimate of the irradiation free V isotope composition of the Moon. Orange squares are from this study, green circles from ref. [16].

the irradiation-free composition of the lunar samples, which has a value of $\delta^{51}V_{Moon} = -1.037 \pm 0.031‰$ ($n = 26$, 2SE), intermediate between chondrites and BSE.

**Vanadium isotope homogeneity of the Moon.** The samples investigated here and elsewhere[16] represent a very diverse set of lunar lithologies covering low and high Ti basalts from different lunar mantle source regions[18], as well as several KREEP-rich (K, Rare Earth Element and Phosphorus) samples corresponding to highly evolved magmas[18]. In agreement with a previous study[16], we conclude that, although V isotope fractionation can be significant at high temperature[19], fractional crystallization did not induce any detectable V isotope fractionation on the Moon as KREEP-rich samples are indistinguishable from mare basalts (Supplementary Table 1). The lack of magmatic V isotope fractionation on the Moon could be related to a lower oxygen fugacity that may have resulted in V occupying a single valence state in lunar magmas, and thus attenuated any redox-driven V isotope fractionation[16]. Given that different types of mare basalts exhibit invariant V isotope compositions that are identical to KREEP-rich rocks (Supplementary Table 1), different regions of the lunar mantle are unlikely to record any detectable V isotope variation. As a result, we infer that the bulk Moon (apart from the very surface that is affected by GCR effects) is homogenous with respect to V isotopes ($\delta^{51}V_{Moon} = -1.037 \pm 0.031‰$; Supplementary Note 2), with an isotopic difference of $\Delta^{51}V_{BSE-Moon} = 0.181 \pm 0.035‰$ (2SE) between the silicate Earth and Moon (Fig. 2; Supplementary Note 2).

**No V isotope effects by evaporation, core formation or late accretion.** The V isotopic difference between Earth and the Moon cannot realistically be explained via kinetic isotope fractionation due to partial volatilization of V during the Moon-forming event. First because V is relatively refractory under both nebular and planetary magma ocean conditions[20,21], and therefore unlikely to have been significantly volatilized. However, even if volatilization induced V isotope effects, then isotopic fractionation during either (i) partial condensation of an originally BSE-like vapor phase or (ii) evaporation of a partially molten proto-Moon[22] would both produce heavy isotope enrichments relative to the BSE, which is opposite to what we observe here for lunar V. If the Moon represents a partial condensate of a protolunar disk[23] that resulted in a light V isotope composition of the Moon relative to Earth, then we would expect similarly refractory elements like Ti and Sr to show similar stable isotope offsets as V, which is not observed[24,25]. Furthermore, equilibrium isotope exchange reactions in the protolunar disk may be expected to produce limited isotope fractionation because V, like Si or Ti, is associated with at least one atom of oxygen (e.g., $VO$, $VO_2$, $V_4O_{10}$) in both the solid and gas phases[26], which limits the potential for significant equilibrium isotope fractionation[27]. The partial vaporization behavior and thermodynamics of V under protolunar disk conditions are unknown, making quantitative assessments of such equilibrium effects very difficult. For Si, the isotopic offset expected between the Earth and the Moon from liquid–vapor separation within the silicate vapor atmosphere[28] is ~3 times smaller than the one reported here for V, and is not observed in natural samples[29]. By analogy to Si, equilibrium isotope fractionation is, therefore, also unlikely to account for the V isotopic offset between Earth and the Moon.

Formation of the lunar core could have sequestered some V, although it would not have left the silicate Moon with a lighter V isotope composition than BSE. The lunar core is indeed expected to be more reduced than the silicate Moon[30], and theoretical considerations of stable V isotope fractionation predict that more

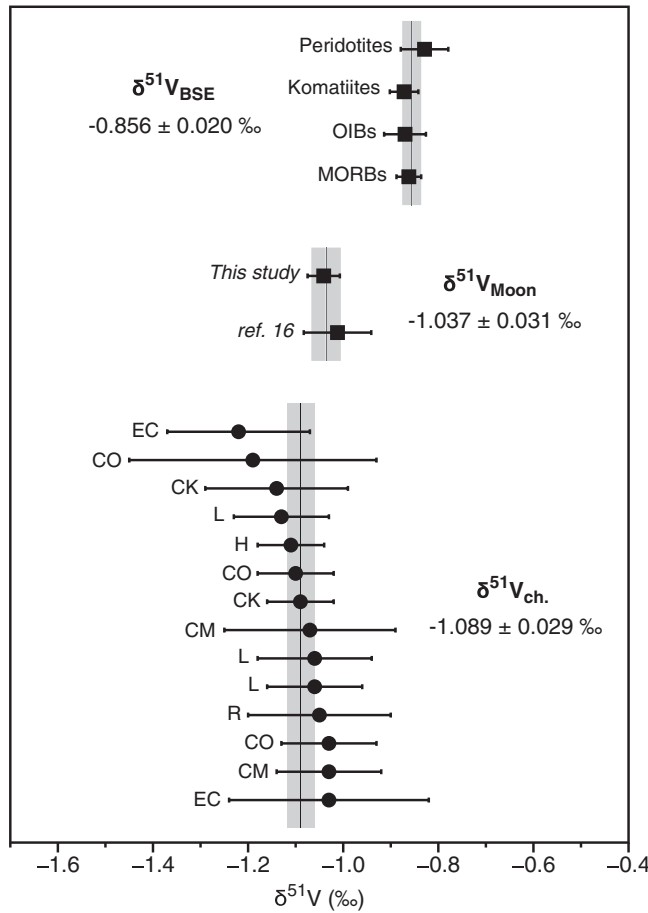

**Fig. 2 Average vanadium isotope compositions for primitive mantle derived terrestrial samples, GCR-corrected lunar samples and GCR-corrected chondrite samples.** Error bars are 2SE for terrestrial samples and the two studies of lunar samples (square markers; Supplementary Note 2). Chondrite data are individual samples with 2 SD error bars (circle markers)[14]. Individual data for terrestrial samples previously compiled[14]. Lunar sample data can be found in Supplementary Table 1. Error-weighted averages and 2SE for each reservoir are shown as vertical gray bars behind each sample group (Supplementary Note 2 and ref. [14]).

reduced forms of V are isotopically lighter[31]. Therefore, lunar core formation would have rendered the silicate Moon heavier than BSE, assuming they both started with a BSE-like isotope composition, which is opposite to the observed difference. In addition, metal-silicate equilibration experiments at 1.5 GPa detected no significant V isotope fractionation[32], implying that low pressure core formation like that of the Moon would not induce any planetary scale V isotope variation. Regarding terrestrial core formation, it is commonly considered that the main phase of metal segregation (pre-Moon formation) readily accounts for the depletion of V in the silicate Earth[33–35], with 40–50% of terrestrial V now residing in the core[35]. This depletion is due to the mildly siderophile nature of V in metal-silicate equilibration experiments over a large range of pressures and temperatures[34–36], which invariably requires large amounts of V to have entered the core throughout Earth accretion. For this reason, it is not likely that post-giant impact terrestrial core formation processes produced the heavy V isotopic signature of the Earth relative to that of the Moon[14].

Late accretion of <2% chondritic material to Earth after the giant impact[37] appears incapable of inducing any change in the V

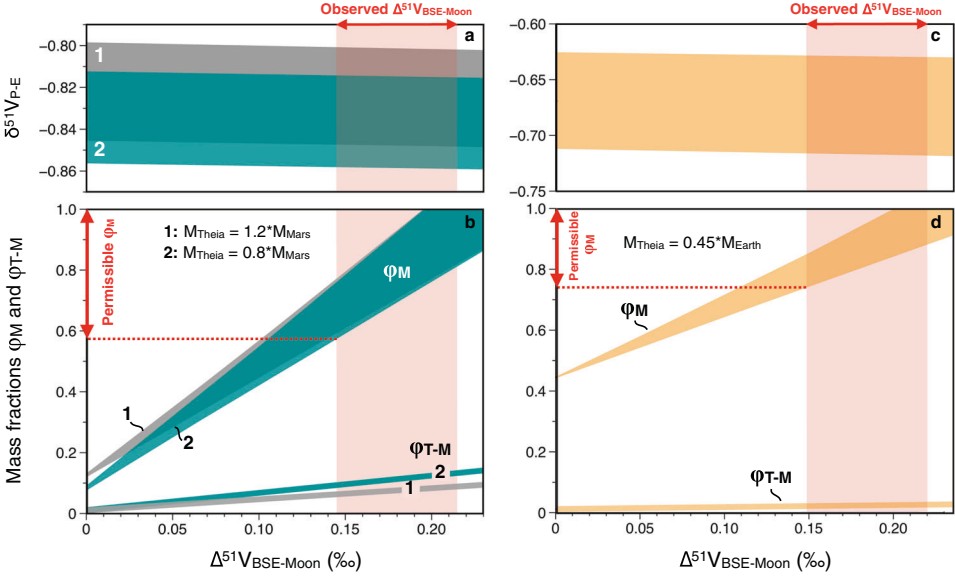

**Fig. 3 Isotope mass balance calculation for giant impact scenarios.** Results of mass balance calculations for giant impacts where Theia's mass ($M_{Theia}$) represents 0.8 or 1.2 times that of Mars ($M_{Mars}$) (**a** and **b**) or 0.45 Earth masses ($M_{earth}$) (**c** and **d**). It is assumed that Theia was chondritic in composition ($\delta^{51}V = -1.089 \pm 0.031$; 2SE) and that the silicate portions of proto-Earth and Theia had identical V concentrations. The mass fractions (i) of the Moon deriving from Theia ($\varphi_M$) and (ii) of Theia that is incorporated into the Moon ($\varphi_{T-M} = \varphi_M * M_{Moon} / M_{Theia}$), are shown as a function of $\Delta^{51}V_{BSE-Moon}$ (Table 1). The mass fraction of the present-day Earth that originates from Theia ($\varphi_E$) is also displayed in (**b** and **d**). Details of these mass balance calculations are reported in the Supplementary Note 3. Given the observed $\Delta^{51}V_{BSE-Moon} = -0.181 \pm 0.035$‰ (95% c.i.), it can be seen that the minimum fraction of Theia in the Moon is ~60%, with any value up to 100% being allowed. The range of typical $\varphi_M$ values derived from the canonical giant impact model is 72–88%. The best estimate for $\varphi_M$ (when $\Delta^{51}V_{BSE-Moon} = -0.181$) ranges from 79% when the impactor is $0.8*M_{Mars}$ to 87% when the impactor is $0.45*M_{Earth}$ (Table 1), in excellent agreement with predictions from the canonical giant impact scenario.

isotope composition of BSE because chondritic material has V concentrations that are similar to BSE[15]. Post giant impact equilibration between Earth and the lunar debris disk or synestia[2,12] is likewise unlikely to account for the observed V isotopic difference. First because the temperatures at which this process would have taken place[2] are so high that stable isotope fractionation should be highly attenuated. Secondly, if V isotopes had been fractionated during such a process, then we would expect to see similar or larger stable isotope effects for many other elements including Mg and Ti, which is not the case[24,38]. Instead, we show that the Earth-Moon V isotopic difference can be readily accounted for by mixing between proto-Earth and a chondritic impactor in the framework of canonical giant impact simulations, in which the Moon is dominated by material from Theia[3,4].

**Canonical giant impact model.** We carried out two-component isotope mixing calculations considering a system with pre-impact (proto-Earth, Theia) and post-impact (Earth, Moon, escaping mass) components[39] (Supplementary Note 3). We found that all sizes of Theia previously investigated by dynamic simulations ($0.8*M_{Mars} \leq M_{Theia} \leq 0.45*M_{Earth}$, where $M_{Mars}$, $M_{Theia}$ and $M_{Earth}$ are the masses of Theia, Mars and Earth, respectively) can be reconciled with the observed $\Delta^{51}V_{BSE-Moon} = 0.181 \pm 0.035$‰ as long as the Moon contains at least ~60% material from Theia (Fig. 3). We note that the calculations assume the silicate portion of Theia to be broadly chondritic, akin to what has been observed for Mars[14]. If Theia had been affected by minor V isotope fractionation during core formation then the results of the mixing calculations would invariably render the fraction of Theia in the Moon higher than what we present here. The minimum amount of Theia material in the Moon is only found when considering the smallest size of Theia combined with the smallest potential V isotopic difference between Earth and Moon, $\Delta^{51}V_{BSE-Moon} = 0.146$‰. Increasing $M_{Theia}$ results in higher fractions of Theia

**Table 1 Outputs of mixing calculations for different sizes of the impactor.**

| $M_{Theia}$ | $0.8*M_{Mars}$ | $1.2*M_{Mars}$ | $0.45*M_{Earth}$ |
|---|---|---|---|
| $\varphi_M$ (%) | 57–100 | 59–100 | 74–100 |
| $\varphi_M$ best estimate (%) | 79 | 80 | 87 |
| $\varphi_E$ (%) | 7.4–7.7 | 11.7–12.0 | 43.8–44.0 |
| $\varphi_{T-M}$ (%) | 9.4–13.3 | 6.4–8.9 | 2.2–2.6 |
| $\delta f_T$ (%) | −93.3 to −86.3 | −89.5 to −79.5 | −59 to −41 |
| $\delta^{51}V_{P-E}$ | −0.859 to −0.815 | −0.848 to −0.801 | −0.717 to −0.629 |

$\varphi_M$: mass fraction of the present-day Moon that originates from Theia.
$\varphi_E$: mass fraction of the present-day Earth that originates from Theia.
$\varphi_{T-M}$: mass fraction of Theia that has been incorporated into the Moon (= $\varphi_M * M_{Moon}$ / $M_{Theia}$).
$\delta f_T$: [$\varphi_E$ /$\varphi_M$ − 1] × 100.
$\delta^{51}V_{P-E}$: V isotopic composition of the proto-Earth.

being incorporated into the Moon (e.g., ≥75% for $M_{Theia}$ = $0.45*M_{Earth}$; Fig. 3 and Table 1). The mass balance required to generate the observed $\Delta^{51}V_{BSE-Moon}$ has been reproduced by the canonical Giant Impact simulations[3,4], large impactor sizes[11], as well as hit-and-run simulations[40]. The latter two model types, however, tend to produce relative fractions of Theia in the Earth and Moon ($\delta f_T \equiv [\varphi_E$ /$\varphi_M - 1] \times 100$, where $\varphi_E$ and $\varphi_M$ are the mass fractions of the silicate portions of the Earth and Moon derived from Theia, respectively) that are relatively similar, thus corresponding to $\delta f_T \sim 0 \pm 30$%[11,40]. Conversely, the canonical giant impact simulations generally produce significantly more negative values[3]. Our models reveal that it is only possible to reproduce $\Delta^{51}V_{Earth-Moon} = 0.181 \pm 0.035$‰ when $-100\% < \delta f_T < -40\%$ (Table 1). This range of $\delta f_T$ is very rarely obtained for fast spinning proto-Earth[10], hit-and-run scenarios[40], and for simulations where Theia is larger than 0.15 $M_{Earth}$[11]. We therefore conclude that, although it is possible to account for the V

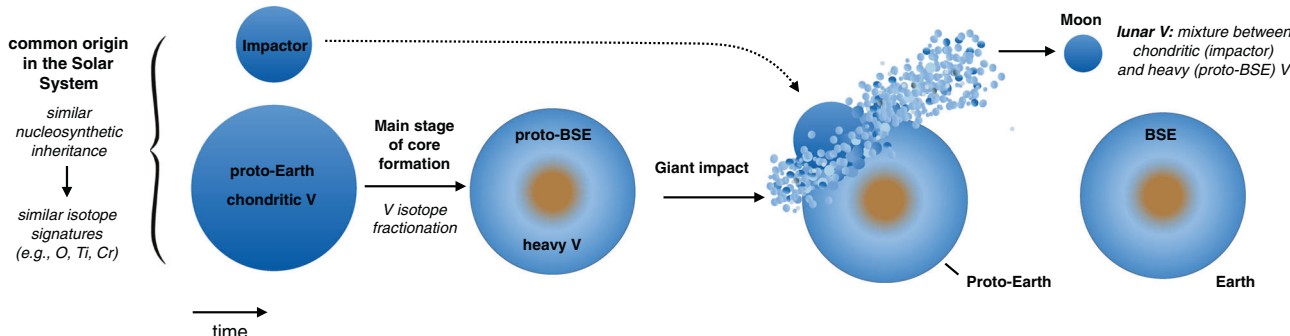

**Fig. 4 Summary cartoon showing our preferred scenario for the origin of the V isotope composition of the Moon.** Theia and proto-Earth would have mainly accreted from a common accretionary reservoir in the inner solar system (which is required to account for their similar nucleosynthetic inheritance)[5, 6, 9, 39]. Subsequently, the proto-Earth experienced its main phase of core-formation at high pressure and high temperature, which caused the V isotope composition of the BSE to be enriched in [51]V relative to its originally chondritic composition[14]. A canonical giant impact between Theia (chondritic V isotope composition) and proto-Earth would then have produced the present-day Earth-Moon system, with ~80% of the lunar accretionary material deriving from the impactor and essentially no difference in the nucleosynthetic assemblages of the Earth and Moon.

isotopic difference between Earth and Moon via multiple different giant impact scenarios, the canonical simulations[3,4] provide a far more robust fit with our observations from V isotopes. Our modeling also places constraints on the fraction of Theia (assumed to have a chondritic composition) that ends up in the Moon ($\varphi_{M-T}$), revealing that only a very small fraction of the total impactor (<14%) is incorporated into the Moon (Table 1). These may be important constraints that could guide future numerical simulations of the canonical giant impact.

**Origin of material in Theia.** The characteristic that sets V isotopes apart from other isotopic tracers of the giant impact is the invariant pre-irradiation V isotope composition of all chondrites (i.e., absence of V nucleosynthetic anomalies). Therefore, the Earth-Moon difference for V isotopes, which was most likely established during core-formation processes predating the Moon-forming giant impact[14], could be explained by Theia containing any type of chondritic material (Fig. 4). As emphasized previously[14], additional experimental investigations of V metal-silicate partitioning at high pressure-temperature conditions, for variable oxygen fugacities and/or chemical compositions will be essential to shed light on V isotopic fractionation processes during terrestrial core formation. However, the fact remains that Earth and the Moon are isotopically very similar for the majority of elements investigated to date[5–7,39]. The explanation of the V isotope difference between the Earth and the Moon in the framework of the canonical giant impact refutes the possibility of post-impact Earth-Moon equilibration processes (e.g., a synestia model or through alternative impact geometries)[2,10–12]. In such scenarios, the V isotope compositions of the Earth and Moon should have been the same, which is not the case. This constraint from V isotopes also implies that the W isotope compositions of the Earth and Moon were most likely not identical in the aftermath of the Giant Impact as recently proposed[41], although even under these circumstances Monte-Carlo simulations still predict that canonical Giant Impact mixing processes would more likely have produced larger W isotope offsets between Earth and the Moon than what is observed[42]. As such, the small W isotope difference between Earth and the Moon invariably implies that the compositions of Theia and proto-Earth, perhaps somewhat fortuitously[43], were more similar for W isotopes than most other differentiated planetary bodies in the solar system. In that sense, one reason why elucidating some of the chemical and isotopic characteristics that resulted from the Moon-forming giant impact has proven so difficult may be that it indeed corresponds to a low-probability event, which cannot be readily predicted from a statistical modeling approach. Lastly, we note that the indistinguishable Si isotope compositions of Earth and Moon[29] would most likely reflect similar planetary formation processes for proto-Earth and Theia[44,45], rather than post impact equilibration between the Earth and Moon[29]. We conclude that the most likely explanation for the Earth-Moon isotopic similarity for other isotope systems than V is that their primordial building blocks originated from a common accretionary reservoir in the inner solar system, therefore comprising broadly similar mixtures of chondritic materials[13]. In particular, enstatite chondrites (and aubrite meteorites) represent our best analogue to Earth's building blocks for many isotope systems such as Ti, O, Cr, and Zr[5,6,9,39,46], and so potentially our best analogue to Theia's composition as well[13,39]. Such a conclusion is also consistent with the recent finding that enstatite-like materials could have been major contributors of terrestrial volatiles[47,48].

## Methods

Samples of both lunar meteorite and Apollo mission rocks were dissolved as either 100 mg chips (meteorites), <40 μm fines, or whole rock powders (Apollo samples) using double distilled concentrated mineral acids such as HF, $HNO_3$, HCl. Vanadium was separated from the sample matrix using a four-step cation/anion exchange chromatography procedure[15,49]. Mass spectrometry to measure V isotope ratios was performed using a Neptune multi-collector inductively coupled plasma mass spectrometer, housed at the Plasma Mass Spectrometry Facility of the Woods Hole Oceanographic Institution (WHOI). Isotope compositions were calculated using standard-sample bracketing with the Alfa Aesar standard that is defined as $\delta^{51}V_{AA} = 0‰$. Each unknown sample was interspersed with a pure V reference solution from BDH Chemicals that has now been measured in four different labs with the identical result of $\delta^{51}V = -1.20‰$[15,49–51]. The mass spectrometer was operated in medium resolution mode, which ensured that all significant isobaric interferences in the mass spectrum (48–53 atomic mass units) were resolved from the isotopes of interest: $^{48}Ti$, $^{49}Ti$, $^{50}V$, $^{51}V$, $^{52}Cr$ and $^{53}Cr$[51,52]. We collected $^{51}V$ using a Faraday cup equipped with a $10^{10}$ Ω resistor, whereas Faraday cups with conventional $10^{11}$ Ω resistors were used for all other masses collected. Samples and standards were measured at a concentration of 800 ng/ml V, which produced an ion beam of ~2 nA on $^{51}V$ and ~0.005 nA on $^{50}V$. Precision and accuracy of the V isotope measurements was assessed by measuring the BDH standard throughout the study ($\delta^{51}V = -1.21 \pm 0.07$; $n = 160$, 2 SD) and by processing USGS reference materials AGV-2, BCR-2 and BHVO-2 that have previously been analyzed by multiple different laboratories[15]. The vanadium isotope compositions and external reproducibility of these measurements during our study were $\delta^{51}V_{AGV-2} = -0.77 \pm 0.07$ ($n = 12$, 2 SD); $\delta^{51}V_{BHVO-2} = -0.87 \pm 0.12$ ($n = 10$, 2 SD); $\delta^{51}V_{BCR-2} = -0.79 \pm 0.08$ ($n = 5$, 2 SD), which are all in excellent agreement with previous studies[15,51,53–55]. Blanks were monitored with each batch of samples and were always <2 ng, which is insignificant compared with the 1000 ng minimum amount of V processed.

Elemental concentrations were determined for all samples using a Thermo-Finnigan iCap quadrupole ICP-MS (Supplementary Table 1), also situated at the WHOI Plasma Facility. Concentrations were calculated via reference to ion beam intensities obtained from a five-point calibration curve constructed from serial dilutions of a gravimetrically-prepared multi-element standard. Drift was

monitored and corrected via normalization to indium intensities. Accuracy and precision were determined to be better than ±7% (SD) based on the correspondence of secondary USGS reference materials AGV-2, BCR-2, and BHVO-2 concentrations determined during the same analytical sessions as the lunar rocks.

## Data availability
The authors declare that all data supporting the findings of this study are available within the paper and in Supplementary Table 1.

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

## Acknowledgements

This study was funded by NASA Emerging Worlds grant NNX16AD36G to S.G.N. We thank NASA-JSC, Tony Irving, and Thorsten Kleine for access to meteorite and Apollo mission samples. US Antarctic meteorite samples are recovered by the Antarctic Search for Meteorites (ANSMET) program, which has been funded by NSF and NASA, and characterized and curated by the Astromaterials Curation Office at NASA Johnson Space Center and the Department of Mineral Sciences of the Smithsonian Institution. J. Blusztajn is thanked for help with mass spectrometry support at WHOI. We also thank Thorsten Kleine, Stephane Le Roux, Rainer Wieler, and Michael Broadley for helpful discussions.

## Author contributions

Study was conceived by S.G.N. Sample processing and V isotope measurements by S.G.N. and M.A. S.G.N. and D.B. interpreted the data and wrote the paper with input from M.A.

## Competing interests

The authors declare no competing interests.
