## [Peer Review File · Nature Communications]

REVIEWER COMMENTS

Reviewer #1 (Remarks to the Author):

In this paper the authors analyzed V isotopic compositions in different lunar samples. In general, they observed resolvable V isotopic differences in all the lunar samples compared with that in the terrestrial samples. Some of the lunar samples suffered galactic cosmic ray bombardment for up to hundreds of million years, which can yield 50V and thereby significantly shift d51V toward negative value. However, the authors applied correlation between Fe/V*exposure age and d51V to eliminate GCE on d51V and estimate the primordial V isotopic composition. As a result, they proposed $d51V_{BSM-BSE} = 0.181 \pm 0.035$ per mil. Several potential scenarios were discussed in this paper, including lunar mantle-core differentiation, partial condensation during Moon accretion, late veneer to the Earth, and post giant impact equilibration between Earth and the lunar debris disk or synestia. However, they ruled out all above scenarios and proposed that the Earth-Moon V isotopic difference can be explained to be incomplete mixing between the proto-Earth and EC-like impactor, in which the Moon inherited the material from the impactor. Overall I truly appreciate their careful work to obtain V isotopic composition in the lunar samples. However, I do have some concerns related to their arguments, including the way to correct GCE and the validity of the model for other chemical and isotopic signatures in the Moon (volatile depletion and MVE isotopic fractionation). This paper will be well qualified for Nature communication if the authors can address my concerns. My questions are listed as follows:

1. I don't think the GCE on d51V for different lunar samples can be simply corrected by d51V vs. Fe/V*CRE age, as Fe is not the only target to generate 50V (targets also include Ti and Cr). To demonstrate the homogeneity of d51V among different lunar samples (soil, high-Ti and low-Ti basalts, and KREEP-rich), their distinct chemical compositions/target abundances have to be taken account combining with their CRE ages to correct d51V. Thus I am wondering that whether the authors can provide chemical composition and estimated shift on d51V corresponding to each sample to firmly address d51V homogeneity in the bulk silicate Moon.
2. The authors analyzed all types of chondrites and shows that different from Ti, Cr, O, and other refractory elements, the values d51V are identical among the chondrites. Given the isotopic similarity of EC and Earth (Moon), the scenario proposed by the authors are reasonable. However, if both proto-Earth and Mars-size impactor originated from a common accretionary reservoir and the present Moon inherit isotopic characteristics of the impactor, additional explanation is required to reproduce "dry" Moon from EC-like composition.
3. Besides V, isotopic distinctions have also been found in other elements, including Cr, Sn (lighter than BSE), K, Rb, Zn, and Ga (heavier than BSE). Due to the rapid solidification of the surface lunar magma ocean (within ~1000 years, Elkins-Tanton et al., 2011, Tang and Young, 2020), the chemical depletion and isotopic fractionation of MVEs (e.g., K, Rb, and Zn) cannot be attributed to LMO evaporation. Instead, such effects must have occurred in the proto-disk right after the giant impact. Therefore, I am wondering that whether the authors can apply their model to explain the chemical and isotopic signatures of other elements in order to address the validity of their model.
4. Can the authors estimate theoretical equilibrium isotopic fractionation factors among V and its oxides to demonstrate whether the V isotopic difference between Earth and Moon is associated with different dominant V species due to oxygen fugacity?
5. This concern is listed here just out of my curiosity: is it possible to measure d51V in iron meteorite to investigate V isotopic fractionation during core formation?

Reviewer #2 (Remarks to the Author):

Review of "Isotopic evidence ... giant impact" by Nielsen et al.

=====

In this manuscript, the authors report new vanadium isotopic measurements on a suite of lunar samples and resolve an isotopic difference between the bulk silicate Moon and bulk silicate Earth of about ~ 0.2 per mil in $^{51}\text{V}/^{50}\text{V}$. They interpret this difference as evidence for the "canonical" giant impact between the proto-Earth and Mars-mass impactor without subsequent isotopic homogenization. If this inference is true, it would be of great significance to the community. I applaud the authors on this intriguing work.

The measurement techniques are state-of-the-art, and I see no reason to doubt the reported isotopic difference between the bulk silicate Moon and bulk silicate Earth.

Regarding the interpretation, however, I have several objections.

First, this type of scenario would have consequences not only for O, Ti, Cr, and V isotopes in lunar rocks (as the authors discuss) but also for other isotopic systems, namely Si and W (which the authors do not discuss). Tungsten isotopes, in particular, are inferred to be identical between the silicate Earth and Moon after the giant impact and to arrive at this circumstance with the scenario that the authors advocate would require a low-probability coincidence (cf. Kruijer and Kleine, 2017 *EPSL* 475 15-24). For this reason, the interpretation advanced by the authors is less convincing given the tungsten isotopic data, which was not addressed in the manuscript.

Second, the very interesting data that the authors report could have a different interpretation, namely, that the liquid-vapor silicate Earth-Moon system did experience isotopic homogenization (as apparently required by Si and W isotopes), but that there was some metal entrainment in the silicate vapor cloud. Indeed, the presence of the lunar core would require some metals to remain suspended during the vapor mixing episode. However, after the cessation of Earth-Moon mixing, these suspended metallic droplets would have been sequestered into cores, at low-pressure on the Moon, and high-pressure on the Earth. Such post-equilibration high-pressure terrestrial core formation could potentially have produced the heavy V isotopic signature reported in the manuscript for the Earth but not Moon. I am not absolutely certain that this sequence occurred. In fact, I am not certain that the high P-T vanadium metal-silicate partitioning data exists to evaluate it. But because I cannot rule it out, there is additional interpretational ambiguity to the scenario advocated in the manuscript.

To summarize: The authors report novel V isotopic data on lunar rocks which display an unexpected difference with the silicate Earth. The authors have squeezed an interesting story out from an element that only has two stable isotopes, and that is laudable. However, both previously reported tungsten isotopic data and interpretational ambiguity regarding high P-T vanadium metal-silicate partitioning leaves me less-than-convinced that the scenario advocated in the manuscript corresponds to the Moon-forming process, and I fear this may also be true for a significant fraction of the lunar origin community. I do not rule out acceptance of this manuscript but would want to see less ambiguity before considering this manuscript suitable for publication.

Reviewer #3 (Remarks to the Author):

This study presents new vanadium (V) isotope data for the Moon, which when corrected for galactic cosmic ray (GCR) exposure yields a $\delta^{51}\text{V}$ isotope composition that is lighter than that of the Bulk Silicate Earth (BSE), and close to that of chondritic meteorites (the so-called, building blocks of the solar system). The light isotope composition of the Moon, relative to the silicate Earth, is not easily explained by volatile loss, fractional crystallisation or core formation (assuming a starting composition similar to that of Earth). These observations can, however, be explained if the heavy $\delta^{51}\text{V}$ isotope composition of the BSE was caused by core formation prior to the Giant-impact that formed the Moon. The Giant-impact itself involved an impactor (Theia) that was chondritic, and 80% of the Moon was formed from this chondritic material. This model is at odds with the Moon being largely derived from the Earth or substantial post-impact equilibration of material from the Earth and Theia.

The paper is clearly written, the data appear to be of the highest quality, the analysis of the data with regard to CRG effects is rigorous, as is the evaluation of the offset with the BSE. Assuming that V is indeed little affected by volatile loss or fractional crystallisation, then this is a potentially very important constraint on the origin of the material from which the Moon was formed, and I strongly recommend publication.

I have two minor comments on the paper:

First, the use of a lunar soil sample in the data used to obtain a lunar $\delta^{51}\text{V}$ isotope composition. If any lunar material is susceptible to contamination with chondritic meteorite material then it is lunar soil (regolith). To clarify this is not contamination by late accretion, but simply being exposed to constant infall at the lunar surface. I would be much more comfortable if the lunar $\delta^{51}\text{V}$ estimate was made without this sample, or at the very least a further estimate made without the inclusion of the soil data.

Second, the interpretation of the V isotope data, that is, that the impactor (Theia) and Earth had a common initial composition that matches that of enstatite chondrites provides an explanation not just for the V data here, but also for a number of other isotopes including Ti, O, Cr, and Zr (long proposed to account for O isotopes). There is, however, one key isotope system that is not given mention, that of Hf-W. If the Moon was largely chondritic, then the W isotope composition of the Moon must be due to lunar core formation, fractionation of Hf/W in the silicate Moon, and subsequent radiogenic ingrowth after the Giant-impact. That is, the Moon was formed while ^{182}Hf was still extant. Rather than being formed late, from material derived from Earth, where the Earth-Moon W isotope differences are explained by late accretion. The model here then is entirely consistent with the study of Thiemens et al. (2019) which proposed that the Hf-W data is best explained by an 'old' age for the Moon, and this merits some mention in this study.

Thiemens et al. Nature Geoscience 12, 696–700 (2019).

Reviewer #1 (Remarks to the Author):

In this paper the authors analyzed V isotopic compositions in different lunar samples. In general, they observed resolvable V isotopic differences in all the lunar samples compared with that in the terrestrial samples. Some of the lunar samples suffered galactic cosmic ray bombardment for up to hundreds of million years, which can yield $\delta^{51}\text{V}$ and thereby significantly shift $\delta^{51}\text{V}$ toward negative value. However, the authors applied correlation between Fe/V *exposure age and $\delta^{51}\text{V}$ to eliminate GCE on $\delta^{51}\text{V}$ and estimate the primordial V isotopic composition. As a result, they proposed $\delta^{51}\text{V}_{\text{BSM-BSE}}=0.181\pm 0.035$ per mil. Several potential scenarios were discussed in this paper, including lunar mantle-core differentiation, partial condensation during Moon accretion, late veneer to the Earth, and post giant impact equilibration between Earth and the lunar debris disk or synestia. However, they ruled out all above scenarios and proposed that the Earth-Moon V isotopic difference can be explained to be incomplete mixing between the proto-Earth and EC-like impactor, in which the Moon inherited the material from the impactor. Overall I truly appreciate their careful work to obtain V isotopic composition in the lunar samples. However, I do have some concerns related to their arguments, including the way to correct GCE and the validity of the model for other chemical and isotopic signatures in the Moon (volatile depletion and MVE isotopic fractionation). This paper will be well qualified for Nature communication if the authors can address my concerns.

We appreciate the positive assessment from reviewer 1 and would like to thank them for their constructive comments. We endeavored to properly address their concerns here below.

My questions are listed as follows:

1. I don't think the GCE on $\delta^{51}\text{V}$ for different lunar samples can be simply corrected by $\delta^{51}\text{V}$ vs. Fe/V *CRE age, as Fe is not the only target to generate 50V (targets also include Ti and Cr). To demonstrate the homogeneity of $\delta^{51}\text{V}$ among different lunar samples (soil, high-Ti and low-Ti basalts, and KREEP-rich), their distinct chemical compositions/target abundances have to be taken account combining with their CRE ages to correct $\delta^{51}\text{V}$. Thus I am wondering that whether the authors can provide chemical composition and estimated shift on $\delta^{51}\text{V}$ corresponding to each sample to firmly address $\delta^{51}\text{V}$ homogeneity in the bulk silicate Moon.

The reviewer is correct that Ti and Cr are alternative potential targets for cosmogenic 50-V production. However, several lines of evidence show that the GCE on $\delta^{51}\text{V}$ for lunar samples can be corrected by $\delta^{51}\text{V}$ vs. Fe/V *CRE age. This question was discussed at length in Hopkins et al (2019) where the authors modeled nuclear cross-sections for V production and burnout and demonstrated that cosmogenic production is primarily affecting V isotope ratios via Ti and/or Fe whereas Cr (and Mn) was shown to be an unlikely target. However, strong correlations between V isotope ratios and CRE age * $[\text{Fe}]/[\text{V}]$ implicate Fe as the primary target element of importance. Plotting sample $\delta^{51}\text{V}$ against exposure age over V concentration (CRE age / $[\text{V}]$) indeed yields a linear correlation, which is markedly stronger than the relationship between $\delta^{51}\text{V}$ and CRE age* $[\text{Ti}]/[\text{V}]$ (see figure below and in Hopkins et al 2019).

Cosmogenic effects from both Ti and Cr are thus likely minimal. In addition, in our recent paper on chondritic and Martian V isotope compositions (Nielsen et al. 2020 GPL), we reached the same conclusion as Hopkins et al 2019, based on the invariance of chondrite V isotope compositions when corrected using Fe, compared with significant scatter when corrected using Ti. The convergence of chondritic V isotope data towards a unique value when corrected for cosmogenic ^{50}V production constitutes another piece of evidence that cosmogenic effects from both Ti and Cr are likely negligible.

In order to make this point very clear in our manuscript, we have added a small paragraph in the supplement section (L39-44), which already discussed the GCR corrections.

2. The authors analyzed all types of chondrites and shows that different from Ti, Cr, O, and other refractory elements, the values $\delta^{51}\text{V}$ are identical among the chondrites. Given the isotopic similarity of EC and Earth (Moon), the scenario proposed by the authors are reasonable. However, if both proto-Earth and Mars-size impactor originated from a common accretionary reservoir and the present Moon inherit isotopic characteristics of the impactor, additional explanation is required to reproduce “dry” Moon from EC-like composition.

We are unsure we fully understand the reviewer's point here. We seem to all agree that the invariant chondritic V isotope composition gives us the unique opportunity to track planetary processes other than source (i.e., nucleosynthetic) effects – here, core formation and binary mixing during giant impact. The proposal that both the proto-Earth and impactor originated from a common accretionary reservoir has been previously advocated in the literature given the isotopic similarity of EC and Earth (Moon) (e.g., Dauphas 2017). Although enstatite chondrites have long been presumed to be devoid of water because they formed in the inner Solar System and are extremely reduced meteorites, Piani et al. (2020) recently analyzed H abundances and isotopic compositions for a suite of enstatite chondrites spanning the full range of thermal

metamorphism degrees and showed that these likely terrestrial building blocks actually contain significant amounts of hydrogen, so that enstatite chondrites could have contributed up to three times the mass of water in the oceans. In this context, the Earth could have acquired a significant fraction of its volatile element inventory prior to the Moon-forming event (as opposed to the concept of late accretion, e.g. Marty, 2012). The concern of the reviewer, as we understand it, is therefore that the Moon might - in this case - be expected to have more water than it does at present. First of all, it is unclear how much water the Moon initially contained. Detections of water on the lunar surface in volcanic rocks, and regolith have led to the proposal that the Moon was actually not as dry as previously thought, although this is debated. It is even considered that the lunar magma ocean and primary crust of the Moon perhaps contained significant amounts of water (Hui et al. 2013). Given that water and volatile escape from the proto lunar disk was most likely inefficient (because it was diffusion-limited), the giant impact hypothesis appears most consistent with a “wet” Moon (Nakajima & Stevenson 2018). Lunar rocks are however severely depleted in moderately volatile elements relative to Earth (e.g., Albarède et al. 2015; Righter 2019) – perhaps this is the “dry” nature of the Moon that the reviewer is referring to? Identifying the cause of this depletion (volatile loss from the disk to the Earth? degassing from the lunar surface?) is probably important for understanding how the Earth– Moon system evolved in the aftermath of the Moon-forming giant impact, but it is unlikely to pertain to V isotopes and we think that this issue is beyond the scope of the present paper (see next comment).

3. Besides V, isotopic distinctions have also been found in other elements, including Cr, Sn (lighter than BSE), K, Rb, Zn, and Ga (heavier than BSE). Due to the rapid solidification of the surface lunar magma ocean (within ~1000 years, Elkins-Tanton et al., 2011, Tang and Young, 2020), the chemical depletion and isotopic fractionation of MVEs (e.g., K, Rb, and Zn) cannot be attributed to LMO evaporation. Instead, such effects must have occurred in the proto-disk right after the giant impact. Therefore, I am wondering that whether the authors can apply their model to explain the chemical and isotopic signatures of other elements in order to address the validity of their model.

We agree with the reviewer that the origin of the chemical depletion and isotopic fractionation of MVEs on the Moon is a fascinating issue, and many scenarios have been proposed: incomplete condensation of a synestia (Lock et al. 2018), Moon formation from a partially condensed and volatile depleted melt (Canup et al. 2015), removal of volatile elements by accretion of the vapor layer onto Earth (Wang et al. 2019), and loss to space from the lunar magma ocean (Day & Moynier 2014; Saxena et al. 2017). Last year, Nie and Dauphas (2019) showed that the Rb, K, Ga, Cu, and Zn of lunar rocks relative to Earth could potentially be explained through (steady state or episodic) viscous drainage of a partially vaporized protolunar disk onto Earth.

We thank the reviewer for suggesting that V isotope insights might help shedding light on this problem. However, as we mention in the text L126-132, V is too refractory for isotope fractionation to have occurred in the partially vaporized protolunar disk. In addition, one would then expect similarly refractory elements like Ti and Sr to show similar stable isotope offsets as V, which is not observed. The V-specific isotope difference between the Earth and Moon is therefore highly unlikely to reflect a volatility-controlled process. In other words, constraining the conditions that led to the chemical depletion and isotopic fractionation of MVEs in lunar rocks is most likely not useful to help understanding the V isotope difference between the Earth and the Moon or vice versa.

4. Can the authors estimate theoretical equilibrium isotopic fractionation factors among V and its oxides to demonstrate whether the V isotopic difference between Earth and Moon is associated with different dominant V species due to oxygen fugacity?

We assume here that the reviewer is referring to stable isotope fractionation processes potentially occurring within a proto-lunar disk with oxygen fugacity gradients. Importantly, even if such gradients existed, these processes appear unlikely to be associated with significant V isotope fractionation. One obvious difficulty in assessing these scenarios is the lack of thermodynamic data constraining the condensation/evaporation behavior of V (and most other elements) at the very high temperatures (presumably >3500 K; Canup et al. 2015; Lock et al. 2018) involved in the formation of the Moon. Although there are no direct calculations of V isotope fractionation under such conditions, we can use previous calculations for Ti in CAIs to obtain a general idea. The magnitude of isotope fractionation between solid and vapor, for a given element, can be estimated from reduced partition function ratios (Schauble 2011), where the magnitude of fractionation is basically controlled by the transformation reaction from vapor to solid phase(s). According to the calculations of Simon et al 2017, Ti would be bonded to oxygen (as TiO and/or TiO₂) in the dominant gas species, therefore limiting differences in reduced partition function ratios between the gas and solid phases and implying negligible equilibrium isotopic fractionation for Ti. Under nebular conditions, V is, like Ti, associated with at least one atom of oxygen (VO, VO₂, V₄O₁₀; Sossi and Fegley 2018), which severely limits the potential for significant equilibrium fractionation of V isotopes. We have now added these arguments to the main text (L132-136).

5. This concern is listed here just out of my curiosity: is it possible to measure d⁵¹V in iron meteorite to investigate V isotopic fractionation during core formation?

This is a very interesting idea! In principle it's possible, but it is difficult for two reasons:

- 1) V concentrations in iron meteorites are typically low (<10ppm).
- 2) CRE ages and Fe/V ratios in iron meteorites are both high, which renders it likely that big GCR corrections will be required to determine the original V isotope composition.

But it is definitely worth doing. Thank you.

Reviewer #2 (Remarks to the Author):

Review of “Isotopic evidence ... giant impact” by Nielsen et al.

=====

In this manuscript, the authors report new vanadium isotopic measurements on a suite of lunar samples and resolve an isotopic difference between the bulk silicate Moon and bulk silicate Earth of about ~0.2 per mil in $51V/50V$. They interpret this difference as evidence for the “canonical” giant impact between the proto-Earth and Mars-mass impactor without subsequent isotopic homogenization. If this inference is true, it would be of great significance to the community. I applaud the authors on this intriguing work. The measurement techniques are state-of-the-art, and I see no reason to doubt the reported isotopic difference between the bulk silicate Moon and bulk silicate Earth.

We thank Reviewer 2 for their positive comments.

Regarding the interpretation, however, I have several objections. First, this type of scenario would have consequences not only for O, Ti, Cr, and V isotopes in lunar rocks (as the authors discuss) but also for other isotopic systems, namely Si and W (which the authors do not discuss). Tungsten isotopes, in particular, are inferred to be identical between the silicate Earth and Moon after the giant impact and to arrive at this circumstance with the scenario that the authors advocate would require a low-probability coincidence (cf. Kruijer and Kleine, 2017 EPSL 475 15-24). For this reason, the interpretation advanced by the authors is less convincing given the tungsten isotopic data, which was not addressed in the manuscript.

We thank the reviewer for this point, which is also discussed by Reviewer 3. Here, the reviewer is correct that W isotopes have been extensively studied for the Moon. It is also correct that some papers infer that the Earth and Moon were identical in the aftermath of the Giant Impact (e.g. Kruijer and Kleine, EPSL, 2017), but it should be noted that the Earth and Moon are, in fact, presently different to each other. This difference was interpreted by Kruijer and Kleine 2017 to reflect a late veneer addition of chondritic material to the silicate Earth, which is certainly possible. However, Thiemens et al, Nature Geoscience, 2019 (which reviewer 3 mentions) presented an alternative view where the Earth and Moon may not have been identical for W isotopes at any point. In other words, the constraints provided by W isotopes against a canonical giant impact are fairly ambiguous.

In our initial manuscript we avoided discussing W isotopes for this very reason. However, we agree with reviewer 2 (and reviewer 3) that it is worth mentioning the two ways of interpreting the W isotope difference between Earth and the Moon, and explaining that our data would render the interpretation by Thiemens et al 2019 more likely (see L207-213 in the present version of the manuscript).

The reviewer also mentions Si isotopes. It is true that the Si isotope compositions of silicate Earth and Moon are within error of each other, which has been interpreted to reflect Earth-Moon equilibration in the aftermath of the giant impact (Armytage et al GCA 2012). However, a series of other papers have, since then, argued that Si isotope compositions of planetary bodies are controlled by volatilization (e.g. Pringle et al PNAS 2014) or nebular equilibration with forsterite (Dauphas et al EPSL 2015). In addition, core formation could also play a significant role in setting planetary Si isotope compositions (e.g. Shahar et al GCA 2011). Thus, it is certainly not straightforward to use Si isotopes as an argument for post giant impact Earth-Moon equilibration, and we therefore do not consider Si isotope systematics to unambiguously refute the hypothesis presented in our contribution. We have added these points in the revised version of the manuscript L207-213.

Second, the very interesting data that the authors report could have a different interpretation, namely, that the liquid-vapor silicate Earth-Moon system did experience isotopic homogenization (as apparently required by Si and W isotopes), but that there was some metal entrainment in the silicate vapor cloud. Indeed, the presence of the lunar core would require some metals to remain suspended during the vapor mixing episode. However, after the cessation of Earth-Moon mixing, these suspended metallic droplets would have been sequestered into cores, at low-pressure on the Moon, and high-pressure on the Earth. Such post-equilibration high-pressure terrestrial core formation could potentially have produced the heavy V isotopic signature reported in the manuscript for the Earth but not Moon. I am not absolutely certain that this sequence occurred. In fact, I am not certain that the high P-T vanadium metal-silicate partitioning data exists to evaluate it. But because I cannot rule it out, there is additional interpretational ambiguity to the scenario advocated in the manuscript.

We thank the reviewer for suggesting this interesting alternative interpretation. We agree that the presence of the lunar core would indicate some metal could have remained suspended during the vapor mixing episode, and that these suspended metallic droplets would have been sequestered into cores, at low-pressure on the Moon, and high-pressure on the Earth. However, for terrestrial core formation, it is commonly considered that the main phase of metal segregation (pre-Moon formation) readily accounts for the depletion of V in the silicate Earth (O'Neill, 1991; Chabot and Agee, 2003; Wade and Wood, 2005), with 40–50 % of terrestrial V now residing in the core (e.g., Wade and Wood, 2005). This is due to the mildly siderophile nature of V in metal-silicate equilibration experiments over a large range of pressures and temperatures (e.g. Chabot and Agee, 2003; Wade and Wood, 2005; Wood et al., 2008), which invariably requires large amounts of V to have entered the core throughout Earth accretion. For this reason, it is not likely that late-stage core formation processes, such as the one proposed by the reviewer, would have caused sufficient V to be sequestered in Earth's core to overall affect the BSE V isotope composition. This is now specified in the manuscript L145-152. In addition, as we already argue in the original manuscript, the lunar core likely did not significantly contribute to modifying the V isotope composition of the silicate Moon because V isotope fractionation factors would have been in the opposite sense to that required to explain the BSE-Moon V isotope difference.

In summary, based on current constraints, it appears that late stage (e.g. sulfide-rich or metal-rich) core formation processes on Earth are unlikely to have removed substantial amounts of V to the core (see Wade and Wood, 2005, Nielsen et al. 2020). However, we completely agree with the reviewer that the lack of high P-T vanadium isotope metal-silicate partitioning data precludes

stronger conclusions to be drawn, and alternative scenarios, such as the one proposed by the reviewer, cannot be firmly be ruled out. L198-201, we added "As emphasized in Nielsen et al. (2020), additional experimental investigations of V metal-silicate partitioning at high pressure-temperature conditions, for variable oxygen fugacities and/or chemical compositions will be essential to shed light on V isotopic fractionation processes during terrestrial core formation." Other scenarios than the canonical giant impact are discussed at length from L125 to L163.

To summarize: The authors report novel V isotopic data on lunar rocks, which display an unexpected difference with the silicate Earth. The authors have squeezed an interesting story out from an element that only has two stable isotopes, and that is laudable. However, both previously reported tungsten isotopic data and interpretational ambiguity regarding high P-T vanadium metal-silicate partitioning leaves me less-than-convinced that the scenario advocated in the manuscript corresponds to the Moon-forming process, and I fear this may also be true for a significant fraction of the lunar origin community. I do not rule out acceptance of this manuscript but would want to see less ambiguity before considering this manuscript suitable for publication.

We understand the reviewer's position and hope that the present changes of the manuscript will address these two concerns.

Reviewer #3 (Remarks to the Author):

This study presents new vanadium (V) isotope data for the Moon, which when corrected for galactic cosmic ray (GCR) exposure yields a $\delta^{51}\text{V}$ isotope composition that is lighter than that of the Bulk Silicate Earth (BSE), and close to that of chondritic meteorites (the so-called, building blocks of the solar system). The light isotope composition of the Moon, relative to the silicate Earth, is not easily explained by volatile loss, fractional crystallisation or core formation (assuming a starting composition similar to that of Earth). These observations can, however, be explained if the heavy $\delta^{51}\text{V}$ isotope composition of the BSE was caused by core formation prior to the Giant-impact that formed the Moon. The Giant-impact itself involved an impactor (Theia) that was chondritic, and 80% of the Moon was formed from this chondritic material. This model is at odds with the Moon being largely derived from the Earth or substantial post-impact equilibration of material from the Earth and Theia.

The paper is clearly written, the data appear to be of the highest quality, the analysis of the data with regard to CRG effects is rigorous, as is the evaluation of the offset with the BSE. Assuming that V is indeed little affected by volatile loss or fractional crystallisation, then this is a potentially very important constraint on the origin of the material from which the Moon was formed, and I strongly recommend publication.

We thank Reviewer 3 for this supportive assessment.

I have two minor comments on the paper:

First, the use of a lunar soil sample in the data used to obtain a lunar $\delta^{51}\text{V}$ isotope composition. If any lunar material is susceptible to contamination with chondritic meteorite material then it is lunar soil (regolith). To clarify this is not contamination by late accretion, but simply being exposed to constant infall at the lunar surface. I would be much more comfortable if the lunar $\delta^{51}\text{V}$ estimate was made without this sample, or at the very least a further estimate made without the inclusion of the soil data.

This is a good point, thank you. It makes no difference to the calculation of the corrected silicate Moon, but there is no question that the reviewer is correct that soil samples should be used with extreme caution when constructing the GCR correction line. Furthermore, lunar soil samples are very heterogeneous and, therefore, it is likely difficult to use CRE ages from other portions of a soil sample to correct for GCR effects on V isotopes, even if chondrite contamination is negligible. We have written a paragraph in the supplement to outline this potential issue (L51-60).

Second, the interpretation of the V isotope data, that is, that the impactor (Theia) and Earth had a common initial composition that matches that of enstatite chondrites provides an explanation not just for the V data here, but also for a number of other isotopes including Ti, O, Cr, and Zr (long proposed to account for O isotopes). There is, however, one key isotope system that is not given mention, that of Hf-W. If the Moon was largely chondritic, then the W isotope composition of

the Moon must be due to lunar core formation, fractionation of Hf/W in the silicate Moon, and subsequent radiogenic ingrowth after the Giant-impact. That is, the Moon was formed while ^{182}Hf was still extant. Rather than being formed late, from material derived from Earth, where the Earth-Moon W isotope differences are explained by late accretion. The model here then is entirely consistent with the study of Thiemens et al. (2019) which proposed that the Hf-W data is best explained by an 'old' age for the Moon, and this merits some mention in this study. Thiemens et al. *Nature Geoscience* 12, 696–700 (2019).

This is a very valid point, which was also commented on by reviewer 2. We now cite the study by Thiemens et al 2019 and briefly discuss constraints from W isotopes on the formation of the Moon. Essentially, we agree with the reviewer 3, although we also note that it is theoretically possible to produce essentially identical W isotope compositions of Earth and Moon via a canonical Giant Impact, as required by V isotopes.

REVIEWER COMMENTS

Reviewer #1 (Remarks to the Author):

Overall I am satisfied with the responses from the authors and recommend to publish this paper on Nature Communication. Sorry that my second comment was not clear. I was wondering about the mechanism proposed in this study to explain water abundance in the Moon. The authors explained that V is refractory element so it did not behave as MVEs or volatiles in the proto-disk. Another issue is the reference of oxygen isotope composition among different objects. Young et al., 2016 does not include the high-precision oxygen data from enstatite chondrite. Please refer Greenwood et al., 2017 (Chemie der Erde).

Reviewer #2 (Remarks to the Author):

Comments on revisions to "Isotopic evidence ... in a canonical giant impact" by Nielsen et al.

The authors have responded quickly and in detail to the concerns raised by the reviewers. I continue to applaud the work done. The reported measurements are valuable to the community. The rebuttal has convinced me that the V isotopic signature on Earth — if due to core formation — is likely a pre-lunar-giant-impact signature. That is helpful and now more clear in the manuscript. I do, however, continue to have a major concern and several minor concerns about how the results are interpreted and presented. These comments are intended to improve the manuscript. If they are adequately addressed, I would recommend the paper for publication.

Major concern

=====

A major concern continues to be the tungsten isotopic record, which has been used to argue *against* the interpretation that the authors advocate in the manuscript. I believe the authors are aware of the state-of-affairs for tungsten, but for the sake of clarity, I reiterate the main points on which I believe everyone should agree: silicate Earth and Moon are both enriched by ~200 ppm in radiogenic ¹⁸²W relative to chondrites (Kleine et al. 2002, Nature 418, 952-955, Touboul et al. 2007 Nature 450, 1206-1209), and are both distinct from meteorites from Mars, Vesta, the aubrites, or any other planetary reservoir. They share a common and distinct tungsten isotopic character.

An order-of-magnitude smaller (~20 ppm) offset *between* the silicate Earth and Moon was expected on the basis of preferential late accretion to the Earth subsequent to lunar formation (see Figure 6 of Halliday, Phil. Trans. R. Soc. A (2008) 366, 4163-4181). This small offset was predicted, sought, and, after laborious laboratory work, resolved by multiple competing groups (Touboul et al. 2015 Nature 520, 530-533, Kruijjer et al. 2015 Nature 520, 534-537), whose conclusions independently converged. This is one of the few predictions in lunar origin geochemistry that has been observed, and the observations have not been disputed in subsequent years. This constraint "is arguably now the most stringent constraint on lunar origin models" (<https://ui.adsabs.harvard.edu/abs/2018AGUFM.V23A..01C/abstract>) according to Robin Canup et al, whom the authors cite.

Thus, Kruijjer and Kleine (2017) conduct mass-balance calculations (an uncontroversial method) using uncontroversial data to determine the a priori probability that silicate Earth and Moon would match with respect to ¹⁸²W to the observed precision in the absence of post-impact-mixing. They find the probability of the scenario that the authors of the current manuscript are advocating as ~1% (see Figure 6c of K+K, 2017 EPSL 475 15-24). If we ignore the corrections due to terrestrial late accretion and take the small E-M offset at face value, the probability may increase to as much as a few percent. Similar considerations led Dauphas et al. (2014, Phil. Trans. R. Soc. A 372: 20130244), when arguing for an inner Solar System origin for the Earth-Moon similarity, to concede that W isotopes would require "a coincidence" to explain. I see nothing wrong with advocating for a low-probability scenario, so long as the authors make clear to the reader that a low-probability coincidence is necessary in the scenario to reproduce the observed EM tungsten

isotopic similarity. I would suggest this caveat be stated in the main text rather than relegated to the supplement.

Alternatively, one way the Earth-Moon system could become isotopically homogeneous with respect to tungsten but not vanadium is if the high fO_2 environment of the post-giant-impact silicate vapor cloud volatilizes tungsten but not vanadium. Tungsten is known to become volatile in the presence of O_2 (e.g., Fegley and Palme, Earth and Planetary Science Letters. 72 (1985) 311-326) and evidence for heterogeneity between the silicate Earth and Moon in refractory elements has been sought (e.g., for Titanium: Zhang, J. et al. 2012 Nature Geosci 5, 251–255) but never found. The vanadium isotopic measurements could be reconciled with the tungsten story if W volatilized and equilibrated but V did not. In any case, if the authors wish to advance inheritance of V isotopes from the impactor, they should make clear that this scenario is unlikely (probability of a few percent per discussion above) to be the explanation for W .

Finally, a note about Thiemens et al. (2019). These authors do report trace element composition measurements and show that the small (~ 20 ppm) offset between the Earth and Moon could potentially derive from radiogenic ingrowth if BSE and BSM started with identical ^{182}W but silicate Moon evolved a higher Hf/W due to lunar core formation. In this way, the *small* offset could still be a chronometer for lunar origin (as reviewer 3 also pointed out). However, please note that this conclusion — even if true — only slightly relaxes the conclusions of Kruijjer and Kleine (2017) who are considering ^{182}W offsets between BSE and BSM of many many hundreds of ppm (see their Figure 6c). Even if Thiemens et al. (2019) are right that ^{182}W is a chronometer for lunar origin, one still needs an explanation for the general Earth-Moon similarity in ^{182}W .

Minor comments

=====

Lines 132-136: The authors have added: "Furthermore, other equilibrium isotope exchange reactions in the protolunar disk would be expected to produce negligible isotope fractionation because V , like Ti , is associated with at least one atom of oxygen (VO , VO_2 , V_4O_{10}) in the gas phase, which severely limits the potential for significant equilibrium isotope fractionation." However, this statement contradicts the behavior of silicon, which is expected to exist as SiO_4^{4-} (+4 valence) in the silicate melt but as SiO (+2 valence) in the vapor, and yet its melt-vapor equilibrium isotopic fractionation is >1 per mil at 2,000 K! (cf. Equation 19 of Pahlevan et al. 2011, Earth and Planetary Science Letters 301, 433–443). If the Earth-Moon V isotopic offset is not due to volatility, then it seems more likely that V was simply too refractory to significantly partition into the vapor. But the magnitude of fractionation between liquid and vapor species is simply unknown, and could easily be significant as in the case of silicon.

Lines 165-191: In these mass-balance calculations, the authors are assuming an undifferentiated (chondritic) Theia. Because the low metal content of the Moon (small core relative to Earth/Mars) requires Theia to be differentiated at the time of the giant impact, then the assumption the authors are making is actually that core formation on Theia produced negligible V isotopic fractionation, somewhat akin to Mars. If so, this should be stated clearly in the main text.

Lines 213-220: The authors state that Earth and Theia derived from the same uniform (?) inner Solar System reservoir. However, what about Mars? Mars is clearly isotopically distinct from the Earth-Moon despite the fact that all dynamical simulations of the accretion of the terrestrial planets have radial mixing and an overlapping source region for the Earth and Mars. Therefore, the authors should make clear that if their advocated scenario is the explanation for Earth-Moon O , Si , Ti , etc., then Mars — the only other sampled terrestrial planet — is anomalous and its accretion history is not described by current N-body simulations.

Reviewer #3 (Remarks to the Author):

All of my major concerns have been addressed, and I would strongly recommend publication.

More than that, the fundamental observation that the Moon and Earth possess a distinct V isotope composition is strongly supported by the data here, and supported by the rigorous assessment in the supplementary information. From my perspective this observation alone justifies the publication of the manuscript, since it brings a key constraint on the origin of the Moon, irrespective of any model favoured by a particular reviewer (or reader).

REVIEWER COMMENTS, responses in red

Reviewer #1 (Remarks to the Author):

Overall I am satisfied with the responses from the authors and recommend to publish this paper on Nature Communication. Sorry that my second comment was not clear. I was wondering about the mechanism proposed in this study to explain water abundance in the Moon. The authors explained that V is refractory element so it did not behave as MVEs or volatiles in the proto-disk. Another issue is the reference of oxygen isotope composition among different objects. Young et al., 2016 does not include the high-precision oxygen data from enstatite chondrite. Please refer Greenwood et al., 2017 (Chemie der Erde).

The requested reference has been added. Thank you.

R. C. Greenwood, T. H. Burbine, M. F. Miller, I. A. Franchi, Melting and differentiation of early-formed asteroids: The perspective from high precision oxygen isotope studies. *Chemie der Erde* 27, 1–43 (2017).

Reviewer #2 (Remarks to the Author):

Comments on revisions to “Isotopic evidence ... in a canonical giant impact” by Nielsen et al.

The authors have responded quickly and in detail to the concerns raised by the reviewers. I continue to applaud the work done. The reported measurements are valuable to the community. The rebuttal has convinced me that the V isotopic signature on Earth — if due to core formation — is likely a pre-lunar-giant-impact signature. That is helpful and now more clear in the manuscript. I do, however, continue to have a major concern and several minor concerns about how the results are interpreted and presented. These comments are intended to improve the manuscript. If they are adequately addressed, I would recommend the paper for publication.

Major concern

=====

A major concern continues to be the tungsten isotopic record, which has been used to argue *against* the interpretation that the authors advocate in the manuscript. I believe the authors are aware of the state-of-affairs for tungsten, but for the sake of clarity, I reiterate the main points on which I believe everyone should agree: silicate Earth and Moon are both enriched by ~200 ppm in radiogenic ^{182}W relative to chondrites (Kleine et al. 2002, *Nature* 418, 952-955, Touboul et al. 2007 *Nature* 450, 1206–1209), and are both distinct from meteorites from Mars, Vesta, the aubrites, or any other planetary reservoir. They share a common and distinct tungsten isotopic character.

An order-of-magnitude smaller (~20 ppm) offset *between* the silicate Earth and Moon was expected on the basis of preferential late accretion to the Earth subsequent to lunar formation

(see Figure 6 of Halliday, Phil. Trans. R. Soc. A (2008) 366, 4163–4181). This small offset was predicted, sought, and, after laborious laboratory work, resolved by multiple competing groups (Touboul et al. 2015 Nature 520, 530–533, Kruijer et al. 2015 Nature 520, 534–537), whose conclusions independently converged. This is one of the few predictions in lunar origin geochemistry that has been observed, and the observations have not been disputed in subsequent years. This constraint “is arguably now the most stringent constraint on lunar origin models” (<https://ui.adsabs.harvard.edu/abs/2018AGUFM.V23A..01C/abstract>) according to Robin Canup et al, whom the authors cite.

Thus, Kruijer and Kleine (2017) conduct mass-balance calculations (an uncontroversial method) using uncontroversial data to determine the a priori probability that silicate Earth and Moon would match with respect to ^{182}W to the observed precision in the absence of post-impact-mixing. They find the probability of the scenario that the authors of the current manuscript are advocating as $\sim 1\%$ (see Figure 6c of K+K, 2017 EPSL 475 15-24). If we ignore the corrections due to terrestrial late accretion and take the small E-M offset at face value, the probability may increase to as much as a few percent. Similar considerations led Dauphas et al. (2014, Phil. Trans. R. Soc. A 372: 20130244), when arguing for an inner Solar System origin for the Earth-Moon similarity, to concede that W isotopes would require “a coincidence” to explain. I see nothing wrong with advocating for a low-probability scenario, so long as the authors make clear to the reader that a low-probability coincidence is necessary in the scenario to reproduce the observed EM tungsten isotopic similarity. I would suggest this caveat be stated in the main text rather than relegated to the supplement.

We thank the reviewer for these correct and relevant comments. We have inserted this important caveat in the main text as requested. Effectively, we agree with the reviewer and concede that, in the framework of the giant impact scenario advocated here, the probability of the small W isotope difference between Earth and Moon is low. This question essentially comes down to the difference in W isotope composition between the silicate portions of proto-Earth and Theia, none of which can be a-priori known. However, we agree that Kruijer and Kleine, 2017 show very nicely that, in the absence of post-impact-mixing, the W isotope similarity between the Earth and Moon must be a coincidence (low probability event).

As the reviewer says below, the assessment presented by Thiemens et al 2019 does make this probability higher, although we agree that it does not affect the overall conclusion of a low-probability event. We have strived to modify the main text to encompass these points, and thank the reviewer for having brought them to our attention.

Alternatively, one way the Earth-Moon system could become isotopically homogeneous with respect to tungsten but not vanadium is if the high $f\text{O}_2$ environment of the post-giant-impact silicate vapor cloud volatilizes tungsten but not vanadium. Tungsten is known to become volatile in the presence of O_2 (e.g., Fegley and Palme, Earth and Planetary Science Letters. 72 (1985) 311-326) and evidence for heterogeneity between the silicate Earth and Moon in refractory elements has been sought (e.g., for Titanium: Zhang, J. et al. 2012 Nature Geosci 5, 251–255) but never found. The vanadium isotopic measurements could be reconciled with the tungsten story if W volatilized and equilibrated but V did not. In any case, if the authors wish to advance inheritance of V isotopes from the impactor, they should make clear that this scenario is unlikely (probability of a few percent per discussion above) to be the explanation for W.

Although this alternative explanation is difficult to entirely rule out, we believe it would be difficult to argue for significant volatilization of W (and not V) during the giant impact. Furthermore, such a process would not be guaranteed to eliminate the inherited W isotope difference between Earth and the Moon (assuming that proto-Earth and Theia were different for W isotopes). We, therefore, think it more appropriate to simply state that the W isotope similarity between Earth and the Moon renders the canonical giant impact a relatively low probability outcome.

Finally, a note about Thiemens et al. (2019). These authors do report trace element composition measurements and show that the small (~20 ppm) offset between the Earth and Moon could potentially derive from radiogenic ingrowth if BSE and BSM started with identical ^{182}W but silicate Moon evolved a higher Hf/W due to lunar core formation. In this way, the *small* offset could still be a chronometer for lunar origin (as reviewer 3 also pointed out). However, please note that this conclusion — even if true — only slightly relaxes the conclusions of Kruijer and Kleine (2017) who are considering ^{182}W offsets between BSE and BSM of many hundreds of ppm (see their Figure 6c). Even if Thiemens et al. (2019) are right that ^{182}W is a chronometer for lunar origin, one still needs an explanation for the general Earth-Moon similarity in ^{182}W .

We agree. See response above, regarding W isotope constraints.

Minor comments

=====
Lines 132-136: The authors have added: “Furthermore, other equilibrium isotope exchange reactions in the protolunar disk would be expected to produce negligible isotope fractionation because V, like Ti, is associated with at least one atom of oxygen (VO, VO₂, V₄O₁₀) in the gas phase, which severely limits the potential for significant equilibrium isotope fractionation.” However, this statement contradicts the behavior of silicon, which is expected to exist as SiO₄⁻⁴ (+4 valence) in the silicate melt but as SiO (+2 valence) in the vapor, and yet its melt-vapor equilibrium isotopic fractionation is >1 per mil at 2,000 K! (cf. Equation 19 of Pahlevan et al. 2011, Earth and Planetary Science Letters 301, 433–443). If the Earth-Moon V isotopic offset is not due to volatility, then it seems more likely that V was simply too refractory to significantly partition into the vapor. But the magnitude of fractionation between liquid and vapor species is simply unknown, and could easily be significant as in the case of silicon.

We thank the reviewer for this pertinent comment. We have now softened this statement given that, as shown for Si isotopes by Pahlevan et al. 2011, non-negligible equilibrium isotope fractionation may still be produced. In the present version of the manuscript, we emphasize that the partial vaporization behavior and thermodynamics of V under protolunar disk conditions are unknown. We then use previous work by Pahlevan et al. 2011 to mention that equilibrium isotope fractionation may theoretically cause small (about three times smaller for Si than observed here for V) isotopic offset between the Earth and the Moon. However, the facts that (i) the magnitude of this offset is markedly smaller than what is observed for V, and (ii) no isotopic offset is observed for Si between the Earth and the Moon (Armytage et al. 2011), indicates that equilibrium isotope fractionation is unlikely to account for the V isotopic offset reported here between Earth and the Moon. The possibility that V was simply too refractory to significantly partition into the vapor is already discussed L129.

Lines 165-191: In these mass-balance calculations, the authors are assuming an undifferentiated (chondritic) Theia. Because the low metal content of the Moon (small core relative to Earth/Mars) requires Theia to be differentiated at the time of the giant impact, then the assumption the authors are making is actually that core formation on Theia produced negligible V isotopic fractionation, somewhat akin to Mars. If so, this should be stated clearly in the main text.

Correct, we have added this assumption to the main text and also outlined the potential consequences of small V isotope fractionation effects during core formation on Theia. Thank you.

Lines 213-220: The authors state that Earth and Theia derived from the same uniform (?) inner Solar System reservoir. However, what about Mars? Mars is clearly isotopically distinct from the Earth-Moon despite the fact that all dynamical simulations of the accretion of the terrestrial planets have radial mixing and an overlapping source region for the Earth and Mars. Therefore, the authors should make clear that if their advocated scenario is the explanation for Earth-Moon O, Si, Ti, etc., then Mars — the only other sampled terrestrial planet — is anomalous and its accretion history is not described by current N-body simulations.

We believe that there is plenty of evidence to suggest that Earth and Mars accreted from different portions of the solar system, regardless of the results from N-body simulations. For example, Earth and Mars accreted over very different time scales, with Mars being fully accreted within only a few million years (Dauphas and Pourmand, 2011) whereas Earth was likely much more protracted (30-50 million years). As such, there are logical reasons why Mars and Earth might be different since, during the major portion of Earth accretion history, Mars was not accumulating more mass. It should also be noted that conventional N-body simulations typically result in a planet at the location of Mars that is much too large, which has led to other models that for instance involve giant planet migration and subsequent reorganization of material in the disk (e.g., Walsh et al. 2011). Although we thank the reviewer for this comment, we feel this discussion is beyond the scope of the present paper.

Reviewer #3 (Remarks to the Author):

All of my major concerns have been addressed, and I would strongly recommend publication.

More than that, the fundamental observation that the Moon and Earth possess a distinct V isotope composition is strongly supported by the data here, and supported by the rigorous assessment in the supplementary information. From my perspective this observation alone justifies the publication of the manuscript, since it brings a key constraint on the origin of the Moon, irrespective of any model favoured by a particular reviewer (or reader).

We thank the reviewer for this supportive feedback.

REVIEWERS' COMMENTS

Reviewer #2 (Remarks to the Author):

The authors have addressed all of my concerns and I am happy to recommend the manuscript for publication.